# Uncovering Subtle Gait Deterioration in People with Early-Stage Multiple Sclerosis Using Inertial Sensors: A 2-Year Multicenter Longitudinal Study

**DOI:** 10.3390/s23229249

**Published:** 2023-11-17

**Authors:** Elisa Gervasoni, Denise Anastasi, Rachele Di Giovanni, Claudio Solaro, Marco Rovaris, Giampaolo Brichetto, Paolo Confalonieri, Andrea Tacchino, Ilaria Carpinella, Davide Cattaneo

**Affiliations:** 1IRCCS Fondazione Don Carlo Gnocchi Onlus, 20148 Milan, Italy; egervasoni@dongnocchi.it (E.G.); danastasi@dongnocchi.it (D.A.); mrovaris@dongnocchi.it (M.R.); davide.cattaneo@unimi.it (D.C.); 2Department of Rehabilitation, Centro di Recupero e Rieducazione Funzionale (CRRF) “Mons. Luigi Novarese”, 13040 Moncrivello, Italy; rachele.digiovanni95@gmail.com; 3Neurology Unit, Galliera Hospital, 16128 Genova, Italy; csolaro@libero.it; 4Italian Multiple Sclerosis Foundation, Scientific Research Area, 16126 Genoa, Italy; giampaolo.brichetto@aism.it (G.B.); andrea.tacchino@aism.it (A.T.); 5IRCCS Foundation “Carlo Besta” Neurological Institute, 20133 Milan, Italy; paolo.confalonieri@istituto-besta.it; 6Department of Physiopathology and Transplants, University of Milan, 20122 Milan, Italy

**Keywords:** gait quality, instrumented gait assessment, rehabilitation, multiple sclerosis, wearable sensors

## Abstract

Limited longitudinal studies have been conducted on gait impairment progression overtime in non-disabled people with multiple sclerosis (PwMS). Therefore, a deeper understanding of gait changes with the progression of the disease is essential. The objective of the present study was to describe changes in gait quality in PwMS with a disease duration ≤ 5 years, and to verify whether a change in gait quality is associated with a change in disability and perception of gait deterioration. We conducted a multicenter prospective cohort study. Fifty-six subjects were assessed at baseline (age: 38.2 ± 10.7 years, Expanded Disability Status Scale (EDSS): 1.5 ± 0.7 points) and after 2 years, participants performed the six-minute walk test (6MWT) wearing inertial sensors. Quality of gait (regularity, symmetry, and instability), disability (EDSS), and walking perception (multiple sclerosis walking scale-12, MSWS-12) were collected. We found no differences on EDSS, 6MWT, and MSWS-12 between baseline and follow-up. A statistically significant correlation between increased EDSS scores and increased gait instability was found in the antero-posterior (AP) direction (r = 0.34, *p* = 0.01). Seventeen subjects (30%) deteriorated (increase of at least 0.5 point at EDSS) over 2 years. A multivariate analysis on deteriorated PwMS showed that changes in gait instability medio-lateral (ML) and stride regularity, and changes in ML gait symmetry were significantly associated with changes in EDSS (F = 7.80 (3,13), *p* = 0.003, R^2^ = 0.56). Moreover, gait changes were associated with a decrease in PwMS perception on stability (*p* < 0.05). Instrumented assessment can detect subtle changes in gait stability, regularity, and symmetry not revealed during EDSS neurological assessment. Moreover, instrumented changes in gait quality impact on subjects’ perception of gait during activities of daily living.

## 1. Introduction

Multiple sclerosis (MS) is a chronic immune-mediated disease characterized by neuronal demyelination and axonal degeneration of the central nervous system [1], often resulting in a progressive disability impacting on gait and the quality of life of people with MS (PwMS) [2].

The disability progression in MS is commonly quantified by an increase in the expanded disability status scale (EDSS) [3]. Previous cross-sectional studies comparing PwMS and healthy subjects have shown gait abnormalities in PwMS [4,5,6,7], revealing that they walk slower, take shorter steps, have a wider base of support, a prolonged double support period, higher gait variability, and poorer dynamic balance, even in the early stage of the disease [4,8,9]. Besides, other cross-sectional studies comparing groups of PwMS with different EDSS scores found that the above walking anomalies are more pronounced in PwMS with higher versus PwMS with lower disability [4,10].

Despite these results, only two studies investigated the longitudinal changes of gait in PwMS with low disability. In details, Dreyer-Alster et al. performed both neurological assessment and gait analysis on eighty-three PwMS in two different time points finding no association between changes in EDSS score and gait changes over time [11], while Galea et al. followed PwMS with EDSS less than three points for one year reporting an overall decrease in balance and gait performances not reflected in clinical status measured with the EDSS [11,12].

In addition to the lack of longitudinal studies, a second issue is that the analysis is usually focused on common spatio-temporal gait parameters (e.g., cadence), which are not always able to capture subtle walking impairments and explain deterioration of locomotion in terms of gait quality and stability [8].

To note, gait deviations are often undetectable during clinical assessment in early-stage PwMS and are not formally assessed by the EDSS [13]. Indeed, EDSS is the gold standard to measure the maximum walking distance covered without assistance or rest, while it is not sensitive to subtle gait deviations that could be present even with an EDSS score below four points [14]. However, these deviations can be measured using wearable sensors providing quality indexes such as gait regularity, gait stability or gait symmetry that have demonstrated significantly reduced, compared to healthy subjects, already in early stages of MS [13,15]. This makes them also promising candidates to comprehensively describe the changes in gait due to disease progression [15]. A recent review by Vienne-Jumeau et al. highlighted the key role of wearable sensors to detect subtle changes in gait quality [16] and several cross-sectional studies have already reported the ability of gait quality indexes to better describe dynamic features of walking in the early phase of MS [8,13,15].

Moreover, it is still unknown whether undetectable changes in gait quality and stability over time are subjectively perceived by non-disabled PwMS. This is of paramount importance to help clinicians to better characterize patient’s impairments, tailoring both pharmacological and rehabilitative treatments over time.

Against this background, our two-year longitudinal cohort study aims at (1) describing changes in gait quality in fully ambulatory, non-disabled PwMS with a diagnosis of relapsing-remitting MS within 5 years, and their association with EDSS changes and (2) verifying whether a change in gait quality is associated with a change in perception of gait deterioration.

## 2. Materials and Methods

### 2.1. Study Design and Participants

Our longitudinal 2-year follow-up study included 80 PwMS in the early stage of the disease recruited from three neurological units of 3 Italian centers [15,17,18]. In the present work, we report data on a subsample of 56 PwMS having a complete clinical and instrumented assessment.

PwMS had to fulfill the following inclusion criteria: clinically defined MS diagnosis [19], EDSS score ≤ 2.5 points corresponding to “no locomotor disability”, disease duration ≤ 5 years, stability of the disease clinically defined as <0.5-point increase in the EDSS disability score over the last 3 months, age > 18 years, and adequate mental capacity to consent. The exclusion criteria were diagnosis of major depression, severe joint and/or bone disorders interfering with balance and gait (based upon clinical judgment), and cardiovascular diseases, or other concomitant neurological diseases.

The Ethics Committee of each involved center approved the study: IRCCS Fondazione Don Carlo Gnocchi Onlus, Milan, Italy (code: 21/2017/CE_FdG/FC/SA), CRRF “Mons. Luigi Novarese”, Moncrivello (VC), Italy (code: AslVC.CRRF.17.03), and Italian Multiple Sclerosis Foundation, Genoa, Italy (code: 026/2018). After having received a full explanation of the study, each subject gave a written informed consent to participate.

### 2.2. Procedures

We assessed participants at baseline and after 2 years. Experienced researchers administered the clinical and instrumented assessments of walking. To ensure standardization, practice assessment sessions were held in the three centers, following written standardized instructions. The same order and timing of testing was maintained during both sessions. The whole assessment battery was performed in a single session allowing participants to rest as needed during the examination. Participants first performed neurologist assessment, then motor assessment (endurance and balance tests), at the end cognitive tests and questionnaires.

### 2.3. Outcome and Outcome Measures

#### 2.3.1. Disability

Experienced neurologists rated the expanded disability status scale (EDSS). The EDSS scale ranges from 0 (normal neurological signs) to 10 (death due to MS). Scores from 1.0 to 4.0 mean normal walking endurance, but reveal impairments in Functional Systems (FS: pyramidal, cerebellar, brainstem, mental, spasticity, sensory, visual, bowel, and bladder) [3].

#### 2.3.2. Walking Endurance

The six-minute walk test (6MWT) is a self-paced submaximal test used to measure walking endurance. It requires the subject to walk as fast and safely as possible back and forth along a 30 m hallway to cover a maximal individual distance within 6 min. The total distance [meter] covered during the test was recorded [20].

#### 2.3.3. Fatigue

The fatigue severity scale (FSS) is a 9-item questionnaire that assesses the impact of fatigue on daily functions. The maximum score is 7, which indicates a high level of fatigue [21].

#### 2.3.4. Instrumented Walking Parameters

Three wireless inertial sensors (MTw, XSens, NL) secured to the lower back, at L5 level, right and left shanks were positioned while performing the 6MWT (see Angelini et al. for further details [22]). We collected sensors signals at a frequency of 75 Hz. Each sensor was composed of a 3D accelerometer (±160 m/s^2^ range), a 3D gyroscope, (±1200 deg/s range), and a 3D magnetometer (±1.5 Gauss).

The portions of acceleration and angular velocity signals related to the 180° turns at the end of each corridor were discarded using the method described by Carpinella et al. [15]. Hence, the following measures were computed from the middle 10 strides of each straight-line walking bout and then averaged over the whole test.

Cadence (stride/min): computed as 60/T_stride_, where T_stride_ is the stride duration (i.e., the time interval between two consecutive heel-strikes of the same foot, estimated following Salarian et al. [23]).Stride regularity (unitless): quantified using the method proposed by Moe-Nilssen & Helbostad [24]. In particular, the normalized autocorrelation function was computed from the trunk acceleration modulus. The first and second peak values of this function, corresponding, respectively, to a time lag equal to step and stride duration, were used to quantify the regularity of consecutive steps and strides (see Angelini et al. [25]). Increasing values, from 0 to 1, indicate higher step and stride regularity. In this study, stride regularity was preferred to step regularity since the latter has been often used as a measure of gait symmetry (see [26,27]).Gait instability (unitless): quantified by the short term Lyapunov exponent (sLyE) computed from the lower back antero-posterior (AP) and medio-lateral (ML) accelerations, as detailed by Caronni et al. [13]. Given that sLyE is affected by data length [28,29,30], each 10-stride steady state walking bout was re-sampled to 1000 frames (10 strides × 100 frames) to maintain equal data length across walking bouts and participants. Hence, sLyE was computed on each time-normalized walking bout and then averaged over the whole test, as proposed by Sloot et al. [31]. Larger values of sLyE mean decreased local dynamic stability, that is decreased ability of the balance control system to deal with small perturbations typically occurring during locomotion, such as internal control errors or external disturbances [28,32].Gait symmetry (%): quantified by the improved Harmonic Ratio (iHR) calculated from trunk AP and ML accelerations [15,18]. In summary, we used a fast discrete Fourier transform to decompose acceleration signals into harmonics. Hence, iHR was computed as the percentage ratio between the sum of the powers of the first 10 in-phase harmonics to the sum of the powers of the first 20 (in-phase and out-of-phase) harmonics. A range from 0 (no symmetry) to 100% (perfect symmetry) was used to describe gait symmetry (see Pasciuto et al. [33]).

The above metrics were chosen as representative of the four gait domains (i.e., rhythm/pace, variability, dynamic balance, and asymmetry) previously proposed in research to describe locomotion in people with neurological diseases, including MS [22,25,27,34,35].

Rhythm/pace domain includes common spatio-temporal parameters (e.g., stride/step time, cadence, stride/step length, swing, stance and double support time [22,25,27,34] which show well documented impairment in PwMS, even at an early stage of the disease [9,10]. Among them, cadence was here selected since, together with stride length, is the key determinant of walking speed, that is considered the “6th vital sign” [36]. Cadence was preferred to stride length based on a previous analysis showing that the present sample of PwMS was characterized by a significantly lower cadence compared to healthy subjects, but comparable stride length [15].

Variability, describing the fluctuations in gait parameters among steps, has demonstrated to be abnormally increased in PwMS [37], and to be a promising early marker of mobility deterioration due to disease progression [4]. In the present study, gait variability domain was represented by the stride regularity parameter. This was preferred to other metrics (i.e., standard deviation or coefficient of variation of spatio-temporal parameters [4,37], since it is less sensitive to the number of considered gait cycles [38,39], and it quantifies the similarity of consecutive strides with a single value rather than 4 or 5 [4,37].

Dynamic balance is one of the first functional impairment due to MS, starting from the very early stages of the disease [18,40] and gradually progressing over time [41] leading to increased risk of falls [42]. Here, gait instability (quantified with AP and ML sLyE) was chosen to represent the dynamic balance domain given that it correlated with clinical scales of balance [13], it is a major contributor of perceived walking ability in daily life [15], it is less influenced by gait speed [13] compared to other stability measures (e.g., root mean square of the acceleration time series [25]), and it is a sensitive metric responsive to rehabilitation effect [43].

Increased gait asymmetry has already demonstrated to be a hallmark of MS, despite the evidence that asymmetric components, reflecting natural functional distinction between lower limbs, are present also during healthy locomotion [44]. Early-stage PwMS already show abnormally high gait asymmetry [15,27], which further increase with disease progression [27]. In the present work, the asymmetry domain was represented by AP and ML iHR chosen as gait symmetry parameters. These metrics were preferred over others (e.g., difference or ratio between the spatio-temporal parameters related to the left and right sides [22,27,45], since iHR quantifies gait symmetry with two values (AP and ML iHR) rather to 4 [22] or 6 [27].

The analysis of the Pearson’s correlation coefficients (r) among the six selected variables (cadence, stride regularity, AP and ML gait instability, AP and ML gait symmetry) revealed no statistically significant associations between them (|r| ≤ 0.25, *p* ≥ 0.060), with the exception of cadence showing a significant correlation with AP gait symmetry (r = 0.28, *p* = 0.037). However, being this *r* value lower than 0.4, the association between the two metrics can be considered low (see [46]). These results suggested that the six metrics chosen for the analysis describe independent aspects of gait.

#### 2.3.5. Perceived Walking Ability

The twelve-multiple sclerosis walking scale (MSWS-12) is a self-rated questionnaire on walking ability. The questionnaire inquires patient’s walking limitations related to MS. Each item scored from 1 to 5; a higher score reflects higher perceived walking difficulties. The items cover different aspects of walking function and quality, such as the ability to walk and climb up and downstairs [47]. To specifically investigate the perceived walking stability and symmetry, we arbitrarily chose, respectively, Item 5 (“How much has your MS limited your balance when standing or walking?”), and Item 11 (“How much has your MS affected how smoothly you walk?”).

### 2.4. Statistical Analysis

For all 56 participants, EDSS and gait changes were calculated as the difference between scores at baseline and scores at the 2-year follow-up. We used the Pearson’s correlation coefficient, adjusted for multiple comparison, to estimate pairwise correlations between changes in gait variables and changes in EDSS and MSWS-12. A linear model was used to assess the multivariate associations between changes in gait variables, as independent predictors, and changes in EDSS as dependent variable. A stepwise procedure (step in R base package) was used to develop a parsimonious model. Beta coefficients, coefficient of determination, residual standard error, and overall F value were reported. Moreover, we used residual plots to check the independence, normality, and constant variance of the model, while the Cook’s distance was used to assess the presence of influential points.

The same procedures were run on the subgroup of participants showing a deterioration in EDSS of at least 0.5 point at follow-up (Deteriorated Group, N = 17), as suggested by Dreyer-Alster et al. [11]. Moreover, all deteriorated participants were divided into subgroups based on their scores on Item 5 and Item 11 of the MSWS-12. In particular, unstable subgroup included subjects reporting increased balance limitations (increased score in Item 5 of the MSWS-12), while Stable group included subjects reporting stable or improved balance (unchanged or decreased score in Item 5 of the MSWS-12). Similarly, asymmetric and symmetric subgroups were composed, respectively, by participants reporting decreased gait symmetry (increased score in Item 11 of the MSWS-12) and stable or improved gait symmetry (unchanged or decreased score in Item 11 of the MSWS-12). Hence, unpaired *t* test was used to compare changes (baseline–follow-up) in instrumented gait variables between unstable and stable, and between asymmetric and symmetric subgroups.

R v3.6.2 was used for data analyses, and *p* < 0.05 was considered statistically significant.

## 3. Results

### 3.1. Clinical Assessment

Table 1 reports the clinical characteristics of the whole sample with participants showing no to mild functional impairments at baseline. Participants were mostly female, and all had a relapsing remitting form of MS. Furthermore, 58% and 28% of them were treated with first- and second-line disease modifying drugs, respectively, while 14% did not take any medication for MS. After two years, 49% of PwMS were treated with first line drugs, 41% with second line drugs, and 10% took no medication. No statistically significant differences in clinical measures (EDSS, 6MWT and MSWS-12) were found between baseline and 2-year follow-up assessment after Bonferroni correction (Table 1).

Table 1 also reports level of fatigue using FSS at baseline and follow-up. Only 21% of PwMS (both at baseline and at follow-up) showed a level of fatigue higher than the cut-off score (>4.75 points) [17] indicating high level of fatigue.

Seventeen subjects (30%) deteriorated (increase of at least 0.5 point at EDSS) over the 2-year period. Figure 1 shows the distribution of EDSS changes scores (Baseline-2-year follow-up). No participants reached the EDSS score of 4 points at follow up (range: 0–3.5 points), meaning that MS did not impact on walking from a neurological point of view, i.e., in terms of walking endurance without assistance and rest.

### 3.2. Instrumented Walking Assessment

No statistically significant differences in instrumented gait variables (cadence, stride regularity, AP and ML gait symmetry, and AP and ML gait instability) were found between baseline and 2-year follow-up assessment after Bonferroni correction (Table 2).

### 3.3. Instrumented Gait Variables and EDSS

Figure 2 (upper panel) displays the associations between changes in EDSS and changes in gait variables and shows a statistically significant bivariate correlation between increased EDSS scores and increased gait instability in the AP direction (r = 0.34, *p* = 0.01). The stepwise procedure used for the multivariate analysis resulted in a model consisting of only one variable, gait instability AP (estimate ± standard error: b = 2.15 ± 0.83, *p* = 0.01. This model was statistically significant, F(1,54) = 6.60, *p* value = 0.01, with a residual standard error of 0.83 points, R^2^ = 0.11, adj R^2^ = 0.1.

Considering the deteriorated group (n = 17) who worsened at the EDSS, no statistically significant differences were found between baseline and 2-year assessment for all instrumented measure after controlling for multiple comparisons (Bonferroni’s correction). Conversely, we found significant correlations between changes in EDSS and changes in stride regularity, gait instability, and gait symmetry in the medio-lateral direction (Table 3). These associations were confirmed by the stepwise procedure in the multivariate model consisting in three variables, changes in gait instability ML: b = 2.42 ± 0.87, *p* = 0.02, Stride regularity: b = −1.81 ± 0.23, *p* = 0.16 and changes in gait symmetry ML: b = −0.03 ± 0.01, *p* = 0.01. This model was statistically significant, F = 7.80 (3.13), *p* value = 0.003, with a residual standard error of 0.49 points, R^2^ = 0.64, adj R^2^ = 0.56.

### 3.4. Instrumented Gait Variables and Perceived Assessment of Gait

Figure 2 (lower panel) shows the correlation between changes in instrumented variables and changes in MSWS-12 total score in the whole sample. Only changes in AP gait instability was significantly associated with changes in MSWS-12 (r = 0.35, *p* = 0.01). This also holds when considering only the deteriorated group (r = 0.73, *p* = 0.001, Table 3).

In the deteriorated group, 35.3% of participants perceived increased balance limitations (unstable group) and showed higher increase in AP gait instability compared to subjects in the stable subgroup (Stable: −0.03 ± 0.1 a.u.; Unstable: −0.16 ± 0.07 a.u.; t = 3.15, *p*-value = 0.007; Figure 3—upper left panel). No statistically significant difference was found in gait instability in the medio-lateral direction (stable: 0.002 ± 0.008 a.u.; unstable: 0.04 ± 0.21 a.u.; t = 038, *p*-value = 0.72, Figure 3—lower left panel). Moreover, 47% of participants in the deteriorated group perceived decreased gait symmetry. Regarding changes in gait symmetry, no difference was found between subjects perceiving an increased gait asymmetry (asymmetric subgroup) compared with subjects perceiving gait symmetry as stable or improved (symmetric group): antero-posterior gait symmetry (symmetric: −1.66 ± 7.49 a.u.; asymmetric: −0.40 ± 7.17 a.u.; t = −0.35, *p*-value = 0.73; Figure 3—upper right panel), medio-lateral gait symmetry (symmetric: −4.05 ±12.25 a.u.; asymmetric = −2.72 ± 10.20 a.u.; t = −0.24, *p*-value = 0.81, Figure 3—lower right panel).

## 4. Discussion

The present longitudinal study aimed at describing the association between disease progression and changes in gait in fully ambulatory, non-disabled PwMS, and verifying whether these changes are perceived by PwMS during the activity of daily living. The main result of this study is that, although changes in EDSS in early-stage PwMS are mostly associated with changes in impairments (i.e., pyramidal or sensory systems) rather than function, instrumented assessment can also reveal subtle changes in a functional task, such as gait. Even if subtle changes in gait quality are difficult to detect during routine neurological assessment through EDSS, they already seem to impact on subjects’ perception of gait ability during activities of daily living.

Overall, we found that more than half of the sample did not show any deterioration of EDSS over two years, suggesting an overall clinical stability of the disease. This result is in line with a previous study by Galea et al. reporting a worsening only in 6 out of 25 PwMS in a similar population in a 1-year longitudinal study [12].

Notably, the EDSS scores of all participants at follow-up still remained below four points, meaning that the changes on this scale only reflect variations at impairment level (e.g., visual and/or bowel and bladder [48]), while walking function, in terms of endurance and level of assistance, is unchanged. By contrasts, the whole sample analysis of instrumented gait variables showed a statistically significant association between changes in the EDSS and AP gait instability indicating that this instrumented gait metric could be a proxy variable able to objectively describe the progression of motor deterioration in PwMS [8]. If corroborated by other larger studies, these results can add information on gait instability, which is strictly related to dynamic balance, indicating that it seems to be more sensitive than other gait metrics (i.e., cadence, regularity, symmetry) to portray subtle gait changes in non-disabled PwMS.

As expected, gait abnormalities were even more evident when considering only participants who showed a worsening in EDSS (deteriorated group). This group showed also statistically significant associations between the change in EDSS and ML gait instability, ML symmetry, and stride regularity. This is in accordance with a previous longitudinal study reporting medio-lateral balance deterioration [12], and with other cross-sectional studies finding that gait becomes less regular and asymmetric as the disease progresses [4,45].

Understanding factors underpinning gait deterioration in the early phase of MS is challenging; however, it is worth speculating on the possible causes. First, it has been recently reported that PwMS in this phase present somatosensory, proprioceptive and vestibular impairments leading to a deterioration of dynamic balance and walking regularity [15,49,50]. Second, motor deterioration can be associated with the axonal loss and asymmetric lesions in the lateral cortico-spinal tract and spinal cord leading to reduced gait symmetry [51]. As suggested by Cofré-Lizama et al. [8], these two factors can impact gait due to the inability of PwMS to control weight shifting during walking leading to imbalance that is increasingly evident with the progression of the disease [52]. However, these speculations need further investigations in future studies.

Taken together, gait instability and, to less extent, asymmetry and regularity are three candidates to reveal early impairments and to monitor gait deterioration and disease progression not detectable on the EDSS [11]. Furthermore, the presence of subtle gait impairments in the early phase of MS highlights the role of preventive rehabilitation aimed at improving dynamic balance and gait symmetry maximizing neuroprotection [53,54], preserving the neurological, physical, and functional reserve, and reducing the progression of disability [55,56,57].

The novelty of the present study was also to investigate if subtle gait disturbances impact on activities of daily living. Although changes in EDSS and gait quality were small, they were already associated with a decrease in perceived gait ability. The present work provides additional evidence with respect to a previous study reporting that a reduced perception of gait quality was associated with worse general gait variables such as gait velocity, stride length, and step width [11]. However, we found that subtle gait deterioration is more closely related to dynamic balance impairment than general gait variables as cadence. This results expand the cross-sectional findings by Carpinella et al. [15,19] showing the association between subtle gait disorders and patient reported walking limitations and suggest that instrumented assessment of gait can be used both to portrait gait disorders and monitor gait changes over time.

Some limitations must be acknowledged regarding the present study. First, a 2-year observation period is too short to detect a firm functional progression on disability. This suggests that gait changes should be modeled over a longer timeframe to comprehensively determine the changes in gait specifically due to MS progression. Second, wearable sensors did not provide information on kinetic (e.g., muscle power) that can be used to assess gait deterioration. Third, although the MSWS-12 is validated to study gait perception in PwMS, no specific data have been published for PwMS with low disability level. Fourth, we did not control for the sleep time the night before the assessment and that could have impacted the performances. Moreover, EDSS reliability and sensitivity could have impacted results on disability changes. Finally, more challenging tasks (e.g., running) could better reveal motor deterioration in early MS.

## 5. Conclusions

The main message of this study is that, although EDSS scores below four do not directly assess the functional deterioration of walking, subtle gait changes can be measured by the objective assessment better describing the perceptions of gait decline in the activity of daily living in the early phase of MS. Our results also suggest that gait instability is relevant and more sensitive than general gait parameters to measure walking deterioration. Taken together, the present findings suggest that the instrumented assessment of gait quality, through easy-to-use wearable inertial sensors, is a good candidate to monitor the progression of gait disorders in this population.

## Figures and Tables

**Figure 1 sensors-23-09249-f001:**
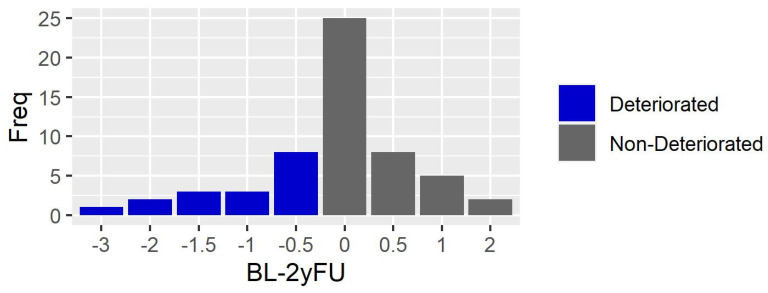
Number of participants (Freq) who deteriorated (blue bars) and not deteriorated (grey bars) over the 2-year follow up, in terms of EDSS change score (BL-2yFU).

**Figure 2 sensors-23-09249-f002:**
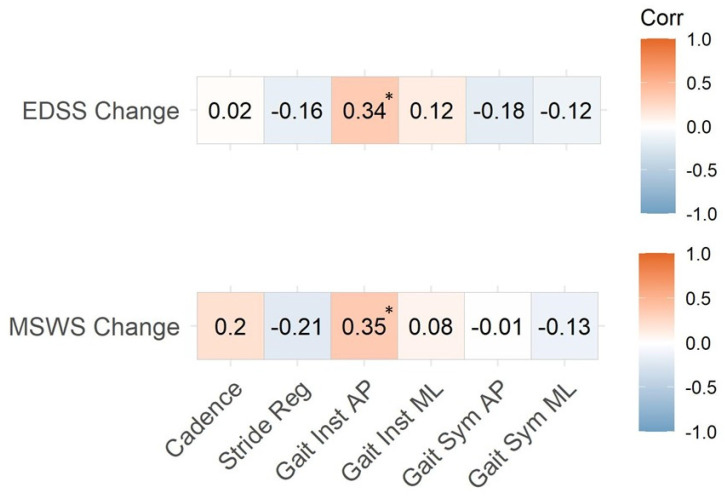
Correlation coefficients between changes in EDSS and MSWS-12 (Baseline vs. 2-year follow-up) and changes in instrumented gait outcome measures on the whole sample (n = 56). EDSS: Expanded Disability Status Scale; MSWS-12: 12-item Multiple Sclerosis Walking Scale; Stride Reg: Stride Regularity; Gait Inst AP: Gait Instability Antero-Posterior; Gait Inst ML: Gait Instability Medio-Lateral; Gait Sym AP: Gait Symmetry Antero-Posterior; Gait Sym ML: Gait Symmetry Medio-Lateral. * denotes statistically significance (*p* < 0.05).

**Figure 3 sensors-23-09249-f003:**
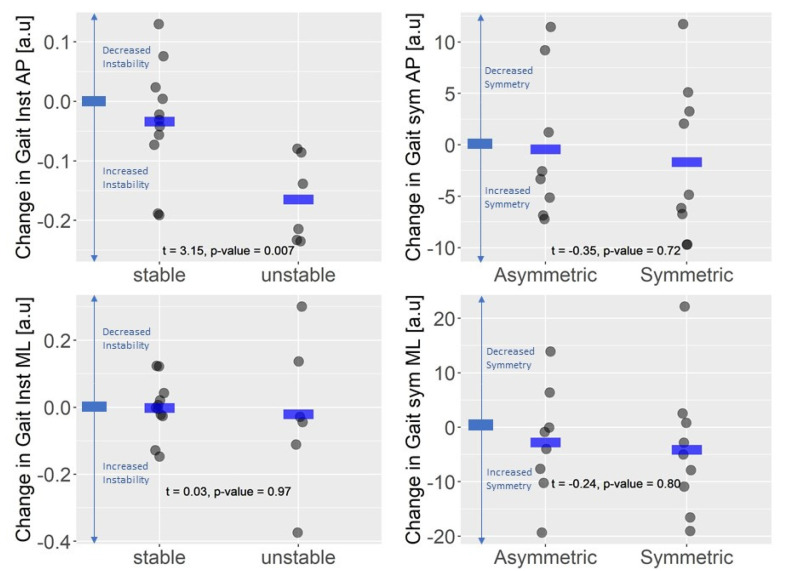
Changes in Gait Instability and Gait Symmetry and changes in perception of walking in the worsened group (n = 17). Gait Inst: Gait Instability; Gait sym: Gait symmetry; AP: antero-posterior direction; ML: medio-lateral direction; Stable: subgroup of subjects reporting stable or improved balance (unchanged or decreased score in Item 5 of the MSWS-12); Unstable: subgroup of subjects reporting deteriorated balance (increased score in Item 5 of the MSWS-12); Asymmetric: subgroup of subjects reporting decreased gait symmetry (increased score in Item 11 of the MSWS-12); Symmetric: subgroup of subjects reporting and stable or improved gait symmetry (unchanged or decreased score in Item 11 of the MSWS-12).

**Table 1 sensors-23-09249-t001:** Baseline and 2-year follow up clinical characteristics of participants (mean ± standard deviation).

Variable	Baseline (n = 56)	2-Year Follow Up (n = 56)	2-Year Follow Up Deteriorated Group (n = 17)	2-Year Follow Up Non-Deteriorated Group (n = 39)
Age (years)	38.2 ± 10.7	40.0 ± 11.0	44.6 ± 9.8	38.4 ± 10.8
Female (n, %)	35, 63%	35, 63%	10, 59%	26, 67%
EDSS (points)	1.5 ± 0.7	1.8 ± 1.0	2.7 ± 0.8	1.4 ± 0.7
Years since diagnosis	2.2 ± 1.8	4.2 ± 1.9	4.1 ± 1.5	4.2 ± 2.1
6MWT (m)	564.0 ± 78.7	574.8 ± 87.6	554.5 ± 108.1	585.5 ± 76.9
MSWS-12 (points)	31.3 ± 14.1	31.4 ± 14.2	38.9 ± 17.0	28.8 ± 12.3
FSS (points)	3.1 ± 1.7	3.2 ± 1.8	3.7 ± 1.8	3.0 ± 1.8

Values are mean ± SD or number, percentage. EDSS: Expanded Disability Status Scale; 6MWT: Six Minute Walk Test; MSWS-12: 12-item Multiple Sclerosis Walking Scale; FSS: Fatigue Severity Scale.

**Table 2 sensors-23-09249-t002:** Baseline and 2-year follow up instrumented gait variables of participants (mean ± standard deviation).

Variable	Baseline (n = 56)	2-Year Follow Up (n = 56)	2-Year Follow Up Deteriorated Group (n = 17)	2-Year Follow Up Non-Deteriorated Group (n = 39)
Cadence (stride/min)	63.4 ± 4.5	64.2 ± 5.6	63.8 ± 7.7	64.7 ± 4.8
Stride Regularity (a.u.)	0.9 ± 0.1	0.9 ± 0.1	0.9 ± 0.1	0.9 ± 0.1
Gait Symmetry AP (a.u.)	82.0 ± 6.3	84.2 ± 5.7	82.7 ± 7.0	84.8 ± 5.0
Gait Symmetry ML (a.u.)	80.2 ± 10.4	83.1 ± 8.9	80.5 ± 11.7	84.1 ± 7.2
Gait Instability AP (a.u.)	0.7 ± 0.1	0.7 ± 0.1	0.8 ± 0.1	0.7 ± 0.1
Gait Instability ML (a.u.)	0.7 ± 0.1	0.8 ± 0.1	0.8 ± 0.1	0.7 ± 0.1

AP: Antero-Posterior; ML: Medio-Lateral.

**Table 3 sensors-23-09249-t003:** Correlation coefficients between changes in EDSS and MSWS-12 (Baseline vs 2-year follow-up) and changes in instrumented outcome measures of the Deteriorated Group (n = 17).

Instrumented Variable	EDSS (n = 17)	MSWS-12 (n = 17)
Cadence	−0.35	0.17
Stride regularity	−0.49 *	−0.24
Gait instability AP	0.36	0.73 *
Gait instability ML	0.52 *	0.20
Gait symmetry AP	−0.45	−0.09
Gait symmetry ML	−0.55 *	−0.07

EDSS: Expanded Disability Status Scale; MSWS-12: 12-Item Multiple Sclerosis Walking Scale; AP; Antero-Posterior; ML: Medio-Lateral. * denotes statistically significance (*p* < 0.05).

## Data Availability

Anonymized data can be available from the corresponding author upon reasonable request.

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
