# Peer review of "Uncovering Subtle Gait Deterioration in People with Early-Stage Multiple Sclerosis Using Inertial Sensors: A 2-Year Multicenter Longitudinal Study"

_sensors, 2023, doi:10.3390/s23229249_

Round 1

Reviewer 1 Report

Comments and Suggestions for Authors

Overall, this is an interesting and well-written study aimed at describing the association between disease progression and changes in gait in fully ambulatory, non-disabled PwMS, and verifying whether these changes are perceived by PwMS during the activity of daily living.

However, while reviewing the study, several questions have arisen in my mind that remain unanswered. Including comments and answers from the authors on these questions could enhance the significance and scientific soundness of the content. My questions are as follows:

1. Lines 25-26 and 131-148

In this study, you utilized Instrumented Walking Parameters to evaluate the quality of gait. These parameters include stride regularity, gait instability, gait symmetry, and cadence. Could you please clarify the rationale behind choosing these specific parameters over others, and support your selection with relevant references?

Could you please confirm that the selected Instrumented Walking Parameters do not complement each other? Please support your assertion with empirical evidence.

2. Lines 69-71: However, these deviations can be measured using wearable sensors providing quality indexes such as gait stability or gait symmetry that are two good candidates to comprehensively determine the changes in gait due to MS progression. [15]

It appears that you assume human gait is a symmetrical motion pattern. However, scientific evidence indicates the presence of an asymmetric component in gait, which reflects a natural functional distinction between limbs [e.g., Sadeghi, H., Allard, P., Prince, F., & Labelle, H. (2000). Symmetry and limb dominance in able-bodied gait: a review. Gait & Posture, 12(1), 34–45. doi:10.1016/s0966-6362(00)00070-9]. If this is accurate, how does it impact the study's objectives, findings, and discussion? Please share your insights, supported by relevant references.

3. Lines 131-132: Stride regularity (unitless): quantified by the second peak of the normalized autocorrelation function computed from the trunk acceleration modulus. [22]

Why did you specifically select the second peak? Please provide a scientific reason and a relevant reference. Reference number 22 does not offer useful information in this regard.

4. Lines 134-136: Gait instability (unitless): quantified by the short term Lyapunov exponent (sLyE) computed from the lower back antero-posterior (AP) and medio-lateral (ML) accelerations, as detailed by Caronni et al. [13].

The Lyapunov exponent (LyE) is a widely used measure for evaluating gait stability. Previous research has indicated that the length of the data affects the resulting LyE value, yet the underlying cause of this variation remains unclear [e.g., Hussain, V. S., Spano, M. L., & Lockhart, T. E. (2020). Effect of data length on time delay and embedding dimension for calculating the Lyapunov exponent in walking. Journal of The Royal Society Interface, 17(168), 20200311. doi:10.1098/rsif.2020.0311]. Does the length of your data impact the resulting LyE? Please comment on this matter and provide relevant references.

5. Lines 288-295: By contrasts, the whole sample analysis of instrumented gait variables showed a statistically significant association between changes in the EDSS and AP gait instability highlighting that this instrumented gait metric can better reflect the adaptability of the neuromuscular system and characterize the progression of motor deterioration in PwMS. [8] If corroborated by other larger studies, these results emphasize that gait instability, which is strictly related to dynamic balance, is more sensitive than other gait metrics (i.e. cadence, regularity, symmetry) to unravel subtle gait changes in non-disabled PwMS.

It appears that you are implying causation based on association. Is this a valid inference? Please share your comments on this matter and substantiate your claim with empirical evidence.

Author Response

We did not receive any new comments from Reviewer 1 during the second round of revisions.

We thank the reviewer.

Reviewer 2 Report

Comments and Suggestions for Authors

Overall this is a well written manuscript. Below are my comments

Introduction

As there are different types of MS, the authors should identify which type of MS (relapse remitting, primary progressive, secondary progressive, progressive remitting) they are specifically interesting in identifying or whether they are trying to identify any diagnosis.

Methodology:

What type of MS were these patients diagnosed with?

When were these assessments taken? Did the researchers control for medication, time of day, etc? Did you account for MS related fatigue on that day? Depending on time of day/time after medication there is evidence that participants will experience fatigue which may influence gait and completion of the 6MWT. 

There is too much missing information in the methodology to correctly assess the results section.

Round 2

Reviewer 2 Report

Comments and Suggestions for Authors

I appreciate the authors responding to my queries and providing excellent information that helps clarify the methodology however, I think the authors must have misunderstood a few of my queries. I'll try to explain them again.

1. On the day that the test was performed did the researchers check whether the participants who were using drugs had taken them the day of the test?

2. I appreciate the authors stating that all data were collected on the same day however, my question is, what was the order of the assessments? (e.g. they first met with neurologist, then completed the fatigue scale, etc) Secondly, did you control for diurnial variations in these patients? There is significant evidence that time of day/sleep the night before significantly impacts performance on functional assessment and subjective reporting of fatigue. Although you can point to changes in EDSS scores as evidence that what you've captured is truly the changes in MS symptoms, evidence suggests that EDSS has issues with reliability and sensitivity (Meyer-Moock, et al, 2014).

The "deteriorated group", can you compare their baseline and 2 year post scores? Is there a significant difference?
